# Measuring Progress in Dictionary Learning for Language Model Interpretability with Board Game Models

**Adam Karvonen**[*]
Independent

**Benjamin Wright**[*]
MIT

**Can Rager**
Independent

**Rico Angell**
UMass, Amherst

**Jannik Brinkmann**
University of Mannheim

**Logan Smith**
Independent

**Claudio Mayrink Verdun**
Harvard University

**David Bau**
Northeastern University

**Samuel Marks**
Northeastern University

## Abstract

What latent features are encoded in language model (LM) representations? Recent work on training sparse autoencoders (SAEs) to disentangle interpretable features in LM representations has shown significant promise. However, evaluating the quality of these SAEs is difficult because we lack a ground-truth collection of interpretable features that we expect good SAEs to recover. We thus propose to measure progress in interpretable dictionary learning by working in the setting of LMs trained on chess and Othello transcripts. These settings carry natural collections of interpretable features—for example, "there is a knight on F3"— which we leverage into *supervised* metrics for SAE quality. To guide progress in interpretable dictionary learning, we introduce a new SAE training technique, *p-annealing*, which improves performance on prior unsupervised metrics as well as our new metrics.[2]

## 1 Introduction

Mechanistic interpretability aims to reverse engineer neural networks into human-understandable components. What, however, should these components be? Recent work has applied Sparse Autoencoders (SAEs) [9, 16], a scalable unsupervised learning method inspired by sparse dictionary learning to find a disentangled representation of language model (LM) internals. However, measuring progress in training SAEs is challenging because we do not know what a gold-standard dictionary would look like, as it is difficult to anticipate which ground-truth features underlie model cognition. Prior work has either attempted to measure SAE quality in toy synthetic settings [57] or relied on various proxies such as sparsity, fidelity of the reconstruction, and LM-assisted autointerpretability [6].

In this work, we explore a setting that lies between toy synthetic data (where all ground-truth features are known; cf. Elhage et al. [24]) and natural language: LMs trained on board game transcripts. This setting allows us to formally specify natural categories of interpretable features, e.g., "there is a knight on `e3`" or "the bishop on `f5` is pinned." We leverage this to introduce two novel metrics for how much of a model's knowledge an SAEs has captured:

---

[*]Equal contribution. Correspondence to: Adam Karvonen <adam.karvonen@gmail.com>.

38th Conference on Neural Information Processing Systems (NeurIPS 2024).

- **Board reconstruction.** Can we reconstruct the state of the game board by interpreting each feature as a classifier for some board configuration?
- **Coverage.** Out of a catalog of researcher-specified candidate features, how many of these candidate features actually appear in the SAE?

These metrics carry the limitation that they are sensitive to researcher preconceptions. Nevertheless, we show that they provide a useful new signal of SAE quality.

Additionally, we introduce *p-annealing*, a novel technique for training SAEs. When training an SAE with *p*-annealing, we use an $L_p$-norm-based sparsity penalty with $p$ ranging from $p = 1$ at the beginning of training (corresponding to a convex minimization problem) to some $p < 1$ (a non-convex objective) by the end of training. We demonstrate that p-annealing improves over prior methods, giving performance on par with the more compute-intensive Gated SAEs from Rajamanoharan et al. [54], as measured both by prior metrics and our new metrics.

Overall, our main contributions are as follows:

1. We train and open-source over 500 SAEs trained on chess and Othello models each.
2. We introduce two new metrics for measuring the quality of SAEs.
3. We introduce *p*-annealing, a novel technique for training SAEs that improves on prior techniques.

## 2   Background

### 2.1   Language models for Othello and chess

In this work, we make use of LMs trained to autoregressively predict transcripts of chess and Othello games. We emphasize that these transcripts only give lists of moves in a standard notation and do not directly expose the board state. Based on behavioral evidence (the high accuracy of the LMs for predicting legal moves) and prior studies of LM representations [36, 47, 33] we infer that the LMs internally model the board state, making them a good testbed setting for studying LM representations.

**Othello.**   Othello is a two-player strategy board game played on an 8x8 grid, with players using black and white discs. Players take turns placing discs on the board, capturing their opponent's discs by bracketing them between their own, causing the captured discs to turn their color. The goal is to have more discs turned to display your color at the end of the game. The game ends if every square on the board is covered or either player cannot make a move.

In our experiments, we use an 8-layer GPT model with 8 attention heads and a $n = 512$ dimensional hidden space, as provided by Li et al. [36]. This model had no prior knowledge of the game or its rules and was trained from scratch on 20 million game transcripts, where each token in the corpus represents a tile on which players place their discs. The game transcripts were synthetically generated by uniformly sampling from the Othello game tree. Thus, the data distribution captures valid move sequences rather than strategic depth. For this model, Li et al. [36] demonstrated the emergence of a world model—an internal representation of the correct board state allowing it to predict the next move—that can be extracted from the model activations using a non-linear probe. Nanda et al. [47] extended this finding, showing that a similar internal representation could be extracted using linear probes, supporting the linear representation hypothesis [46].

**Chess.**   Othello makes a natural testbed for studying emergent internal representations since the game tree is far too large to memorize. However, the rules and state are not particularly complex. Therefore, we also consider a language model trained on chess game transcripts with identical architecture, provided by Karvonen [33]. The model again had no prior knowledge of chess and was trained from scratch on 16 million human games from the Lichess chess game database [38]. The input to the model is a string in the Portable Game Notation (PGN) format (e.g., "`1. e4 e5 2. Nf3 ...`"). The model predicts a legal move in 99.8 % of cases and, similar to Othello, it has an internal representation of the board state that can be extracted from the internal activations using a linear probe [33].

---

[2]Code, models, and data are available at `https://github.com/adamkarvonen/SAE_BoardGameEval`

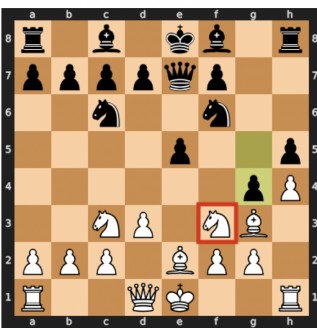 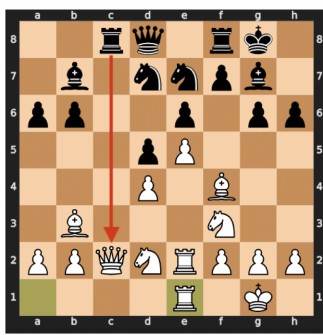 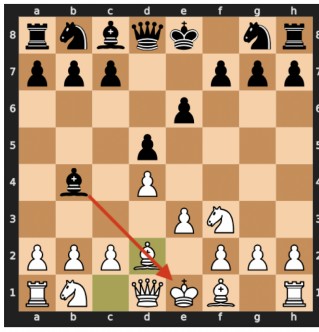

Figure 1: We find SAE features that detect interpretable board state properties (BSP) with high precision (i.e., above 0.95). This figure illustrates three distinct chessboard states, each an example of a BSP associated with a high activation of a particular SAE feature. Left: A **board state detector** identifies a knight on square f3, owned by the player to move. Middle: A **rook threat detector** indicates an immediate threat posed by a rook to a queen regardless of location and piece threatened. Right: A **pin detector** recognizes moves that resolve a check on a diagonal by creating a pin, again, regardless of location and piece pinned.

## 2.2 Sparse autoencoders

Given a dataset $\mathcal{D}$ of vectors $\mathbf{x} \in \mathbb{R}^d$, a sparse autoencoder (SAE) is trained to produce an approximation

$$\mathbf{x} \approx \sum_i^{d_{\text{SAE}}} f_i(\mathbf{x})\mathbf{d}_i + \mathbf{b} \tag{1}$$

as a sparse linear combination of *features*. Here, the *feature vectors* $\mathbf{d}_i \in \mathbb{R}^d$ are unit vectors, the *feature activations* $f_i(\mathbf{x}) \geq 0$ are a sparse set of coefficients, and $\mathbf{b} \in \mathbb{R}^d$ is a bias term. Concretely, an SAE is a neural network with an encoder-decoder architecture, where the encoder maps $\mathbf{x}$ to the vector $\mathbf{f} = [f_1(\mathbf{x}) \quad \ldots \quad f_{d_{\text{SAE}}}(\mathbf{x})]$ of feature activations, and the decoder maps $\mathbf{f}$ to an approximate reconstruction of $\mathbf{x}$.

In this paper, we train SAEs on datasets consisting of activations extracted from the residual stream after the sixth layer for both the chess and Othello models. At these layers, linear probes trained with logistic regression were accurate for classifying a variety of properties of the game board [33, 47]. For training SAEs, we employ a variety of SAE architectures and training algorithms, as detailed in Section 4.

## 3 Measuring autoencoder quality for chess and Othello models

Many of the features learned by our SAEs reflect uninteresting, surface-level properties of the input, such as the presence of certain tokens. However, upon inspection, we additionally find many SAE features which seem to reflect a latent model of the board state, e.g., features that reflect the presence of certain pieces on particular squares, squares that are legal to play on, and strategy-relevant properties like the presence of a pin in chess (Figures 1 and 6).

Fortunately, in the setting of board games, we can formally specify certain classes of these interesting features, allowing us to more rigorously detect them and use them to understand our SAEs. In Section 3.1, we specify certain classes of interesting game board properties. Then, in Section 3.2, we leverage these classes into two metrics of SAE quality.

### 3.1 Board state properties in chess and Othello models

We formalize a *board state property* (BSP) to be a function $g : \{\text{game board}\} \rightarrow \{0, 1\}$. In this work, we will consider the following interpretable classes of BSPs:

- $\mathcal{G}_{\text{board state}}$ contains BSPs which classify the presence of a piece at a specific board square, where the board consists of $8 \times 8$ squares in both games. For chess, we consider the full board for the twelve

distinct piece types (white king, white queen, ..., black king), giving a total of $8 \times 8 \times 12$ BSPs. For Othello, we consider the full board for the two distinct piece types (black and white), yielding $8 \times 8 \times 2$ BSPs.

- $\mathcal{G}_{\text{strategy}}$ consists of BSPs relevant for predicting legal moves and playing strategically in chess, such as a pin detector. They were selected by the authors based on domain knowledge and prior interpretability work on the chess model AlphaZero [45]. We provide a full list of strategy BSPs in Table 3 in the Appendix. Because our Othello model was trained to play random legal moves, we do not consider strategy BSPs for Othello.

## 3.2 Measuring SAE quality with board state properties

In this section, we introduce two metrics of SAE quality: coverage and board reconstruction.

**Coverage.** Given a collection $\mathcal{G}$ of BSPs, our coverage metric quantifies the extent to which an SAE has identified features that coincide with the BSPs in $\mathcal{G}$. In more detail, suppose that $f_i$ is an SAE feature and $t \in [0, 1]$ is a threshold, we define the function

$$\phi_{f_i,t}(\mathbf{x}) = \mathbb{I}\left[f_i(\mathbf{x}) > t \cdot f_i^{\max}\right] \tag{2}$$

where $f_i^{\max}$ is (an empirical estimate of) $\max_{\mathbf{x} \sim \mathcal{D}} f_i(\mathbf{x})$, the maximum value that $f_i$ takes over the dataset $\mathcal{D}$ of activations extracted from our model, and $\mathbb{I}$ is the indicator function. We interpret $\phi_{f_i,t}$ as a binary classifier; intuitively, it corresponds to binarizing the activations of $f_i$ into "on" vs. "off" at some fraction $t$ of the maximum value of $f_i$ on $\mathcal{D}$. Given some BSP $g \in \mathcal{G}$, let $F_1(\phi_{f_i,t}; g) \in [0, 1]$ denote the F1-score for $\phi_{f_i,t}$ classifying $g$. Then we define the *coverage* of an SAE with features $\{f_i\}$ relative to a set of BSPs $\mathcal{G}$ to be

$$\text{Cov}(\{f_i\}, \mathcal{G}) := \frac{1}{|\mathcal{G}|} \sum_{g \in \mathcal{G}} \max_t \max_{f_i} F_1(\phi_{f_i,t}; g). \tag{3}$$

In other words, we take, for each $g \in \mathcal{G}$, the $F_1$-score of the feature that best serves as a classifier for $g$, and then take the mean of these maximal $F_1$-scores. An SAE receives a coverage score of 1 if, for each BSP $g \in \mathcal{G}$, it has some feature that is a perfect classifier for $g$. Since Cov depends on the choice of threshold $t$, we sweep over $t \in \{0, 0.1, 0.2, \ldots, 0.9\}$ and take the best coverage score; typically this best $t$ is in $\{0, 0.1, 0.2\}$.

**Board reconstruction.** Again, let $\mathcal{G}$ be a set of BSPs. Intuitively, the idea of our board reconstruction metric is that, for a sufficiently good SAE, there should be a simple, human-interpretable way to recover the state of the board from the profile of feature activations $\{f_i(\mathbf{x})\}$ on an activation $\mathbf{x} \in \mathbb{R}^d$. Here, the activation $\mathbf{x}$ was extracted after the post-MLP residual connection in layer 6.

We will base our board reconstruction metric around the following human-interpretable way of recovering a board state from a feature activation profile; we emphasize that different ways of recovering boards from feature activations may lead to qualitatively different results. This recovery rule is based on the assumption that interpretable SAE features tend to be *high precision* for some subset of BSPs, in line with Templeton et al. [58]. For example, features that classify common configurations of pieces are high precision (but not necessarily high recall) for multiple BSPs. We use a consistent dataset of 1000 games as our training set $\mathcal{D}_{\text{train}}$ for identifying high-precision features across all Board State Properties (BSPs). An additional, separate set of 1000 games serves as our test set $\mathcal{D}_{\text{test}}$. Using the training set $\mathcal{D}_{\text{train}}$, we identify, for each SAE feature $f_i$, all of the BSPs $g \in \mathcal{G}$ for which $\phi_{f_i,t}$ is a *high precision* (of at least 0.95) classifier. Then for each $g \in \mathcal{G}$ our prediction rule is

$$\mathcal{P}_g(\{f_i(\mathbf{x})\}) = \begin{cases} 1, & \text{if } \phi_{f_i,t}(\mathbf{x}) = 1 \text{ for any } f_i \text{ which} \\ & \quad \text{is high precision for } g \text{ on } \mathcal{D}_{\text{train}} \\ 0, & \text{otherwise.} \end{cases} \tag{4}$$

Let $F_1(\mathcal{P}(\{f_i(\mathbf{x})\}); \mathbf{b})$ denote the $F_1$-score for a given board state $\mathbf{b}$, where $\mathcal{P}(\{f_i(\mathbf{x})\}) = \{\mathcal{P}_g(\{f_i(\mathbf{x})\})\}_{g \in \mathcal{G}}$ represents the full predicted board (containing predictions for all 64 squares) obtained from the SAE activations.[3]

---

[3] We do not score empty squares. Thus, the reconstruction score would be zero if no high precision features are active.

Then, the average $F_1$-score over all board states in the test dataset $\mathcal{D}_{\text{test}}$ can be calculated as

$$\text{Rec}(\{x_i\}, \mathcal{D}_{\text{test}}) = \frac{1}{|\mathcal{D}_{\text{test}}|} \sum_{\mathbf{x} \in \mathcal{D}_{\text{test}}} \max_t F_1(\mathcal{P}(\{f_i(\mathbf{x})\}); \mathbf{b}). \tag{5}$$

## 4 Training methodologies for SAEs

In our experiments, we investigate four methods for training SAEs, as explained in this section. These are given by two autoencoder architectures and two training methodologies—one with $p$-annealing and one without $p$-annealing—for each architecture. Our SAEs are available at `https://huggingface.co/adamkarvonen/chess_saes/tree/main` (chess) and `https://huggingface.co/adamkarvonen/othello_saes/tree/main` (Othello).

### 4.1 Standard SAEs

Let $n$ be the dimension of the model's residual stream activations that are input to the autoencoder, $m$ the autoencoder hidden dimension, and $s$ the dataset size. Our baseline "standard" SAE architecture, as introduced in Bricken et al. [9] is defined by encoder weights $W_e \in \mathbb{R}^{m \times n}$, decoder weights $W_d \in \mathbb{R}^{n \times m}$ with columns constrained to have a $L_2$-norm of 1, and biases $b_e \in \mathbb{R}^m$, $b_d \in \mathbb{R}^n$. Given an input $\mathbf{x} \in \mathbb{R}^n$, the SAE computes

$$\mathbf{f}(\mathbf{x}) = \text{ReLU}(W_e(\mathbf{x} - \mathbf{b}_d) + \mathbf{b}_e) \tag{6}$$
$$\hat{\mathbf{x}} = W_d \, \mathbf{f}(\mathbf{x}) + \mathbf{b}_d \tag{7}$$

where $\mathbf{f}(\mathbf{x})$ is the vector of feature activations, and $\hat{\mathbf{x}}$ is the reconstruction. For a standard SAE, our baseline training method is as implemented in the open-source `dictionary_learning` repository [43], optimizing the loss

$$\mathcal{L}_{\text{standard}} = \mathbb{E}_{\mathbf{x} \sim \mathcal{D}_{\text{train}}} \Big[ \|\mathbf{x} - \hat{\mathbf{x}}\|_2 + \lambda \|\mathbf{f}(\mathbf{x})\|_1 \Big]. \tag{8}$$

for some hyperparameter $\lambda > 0$ controlling sparsity.

### 4.2 Gated SAEs

The $L_1$ penalty used in the original training method encourages feature activations to be smaller than they would be for optimal reconstruction [62]. To address this, Rajamanoharan et al. [54] introduced a modification to the original SAE architecture that separates the selection of dictionary elements to use in a reconstruction and estimating the coefficients of these dictionary elements. This results in the following gated architecture:

$$\pi_{\text{gate}}(\mathbf{x}) := W_{\text{gate}}(\mathbf{x} - \mathbf{b}_d) + \mathbf{b}_{\text{gate}} \tag{9}$$

$$\tilde{\mathbf{f}}(\mathbf{x}) := \mathbb{I}\left[\pi_{\text{gate}}(\mathbf{x}) > 0\right] \odot \text{ReLU}(W_{\text{mag}}(\mathbf{x} - \mathbf{b}_d) + \mathbf{b}_{\text{mag}}) \tag{10}$$

$$\hat{x}(\tilde{\mathbf{f}}(\mathbf{x})) = W_d \tilde{\mathbf{f}}(\mathbf{x}) + \mathbf{b}_d \tag{11}$$

where $\mathbb{I}[\cdot > 0]$ is the Heaviside step function and $\odot$ denotes elementwise multiplication. Then, the loss function uses $\hat{x}_{\text{frozen}}$, a frozen copy of the decoder:

$$\begin{aligned} \mathcal{L}_{\text{gated}} := \mathbb{E}_{\mathbf{x} \sim \mathcal{D}_{\text{train}}} \Big[ &\|\mathbf{x} - \hat{x}(\tilde{\mathbf{f}}(\mathbf{x}))\|_2^2 \\ &+ \lambda \|\text{ReLU}(\pi_{\text{gate}}(\mathbf{x}))\|_1 \\ &+ \|\mathbf{x} - \hat{x}_{\text{frozen}}(\text{ReLU}(\pi_{\text{gate}}(\mathbf{x})))\|_2^2 \Big]. \end{aligned} \tag{12}$$

### 4.3 $p$-Annealing

Fundamentally, an $L_1$ penalty has been used to induce sparsity in SAE features because it serves as a convex relaxation of the true sparsity measure, the $L_0$-norm. The $L_1$-norm is the convex hull of the $L_0$-norm, making it a tractable alternative for promoting sparsity [63]. However, the proxy loss function is not the same as directly optimizing for sparsity, leading to issues such as feature *shrinkage* [62] and potentially less sparse learned features. Unfortunately, the $L_0$-norm is non-differentiable and directly minimizing it is an NP-hard problem [48, 18], rendering it impractical for training.

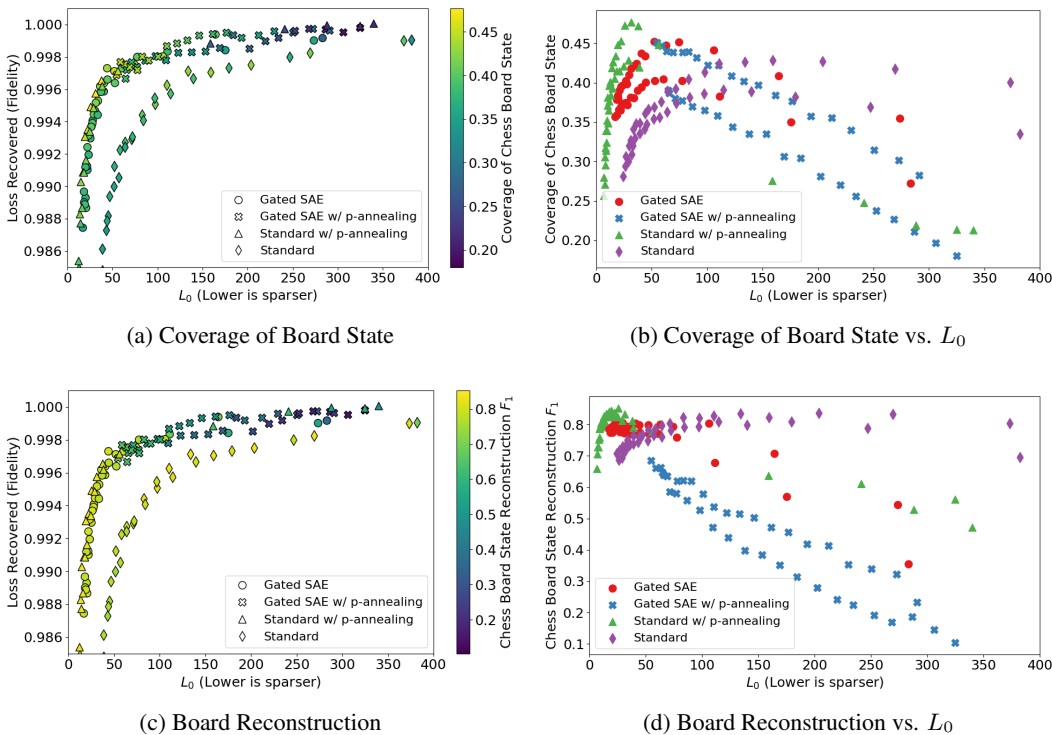

(a) Coverage of Board State

(b) Coverage of Board State vs. $L_0$

(c) Board Reconstruction

(d) Board Reconstruction vs. $L_0$

Figure 2: Comparison of the coverage and board reconstruction metrics for chess SAE quality on $\mathcal{G}_{\text{board state}}$. The coverage score reports the mean F1 scores over BSPs. The top row corresponds to coverage, and the bottom row corresponds to board reconstruction. The left column contains a scatterplot of loss recovered vs. $L_0$, with the scheme color corresponding to the coverage score and each point representing different hyperparameters. We differentiate between SAE training methods with shapes.

In this work, we propose the use of nonconvex $L_p^p$-minimization, with $p < 1$, as an alternative to the standard $L_1$ minimization in sparse autoencoders (SAEs). This approach has been successfully employed in compressive sensing and sparse recovery to achieve even sparser representations [11, 61, 64, 60]. To perform this optimization, we introduce a method called *p-annealing* for training SAEs, based on the compressive sensing technique called $p$-continuation [66]. The key idea is to start with convex $L_1$-minimization through setting $p = 1$ and progressively decrease the value of $p$ during training, resulting in closer approximations of the true sparsity measure, $L_0$, as $p$ approaches 0. We define the sparsity penalty for each batch $x$ as a function of the current training step $s$:

$$\mathcal{L}_{\text{sparse}}(\mathbf{x}, s) = \lambda_s \|\mathbf{f}(\mathbf{x})\|_{p_s}^{p_s} = \lambda_s \sum_i f_i(\mathbf{x})^{p_s} \tag{13}$$

In other words, the sparsity penalty will be a scaled $L_p^p$ norm of the SAE feature activations, with $p$ decreasing over time. At $p = 1$, the $L_p^p$ norm is equal to the $L_1$ norm, and as $p \to 0$, the $L_p^p$ norm limits to the $L_0$-norm, as $\lim_{p \to 0} \sum_i f_i(\mathbf{x})^p = \sum_i f_i(\mathbf{x})^0$.

The purpose of annealing $p$ from $1 \to 0$ instead of starting from a fixed, low value for $p$ is that the lower the $p$, the more concave (non-convex) the $L_p^p$ norm is, increasing the likelihood of the training process getting stuck in local optima, which we have observed in initial experiments. Therefore, we aim to first arrive at a region of an optimum using the easier-to-train $L_1$ penalty and then gradually shift the loss function. This manifests as keeping $p = 1$ for a certain number of steps and then starting decreasing $p$ linearly down to $p_{\text{end}} > 0$ at the end of training. We set $p_{\text{end}} = 0.2$.

**Coefficient Annealing.** Changing the value of $p$ changes the scale of the $L_p^p$ norm. Without also adapting the coefficient $\lambda$, the strength of the sparsity penalty would vary too wildly across training. Empirically, we found that keeping a constant $\lambda$ would lead to far too weak of a sparsity penalty for

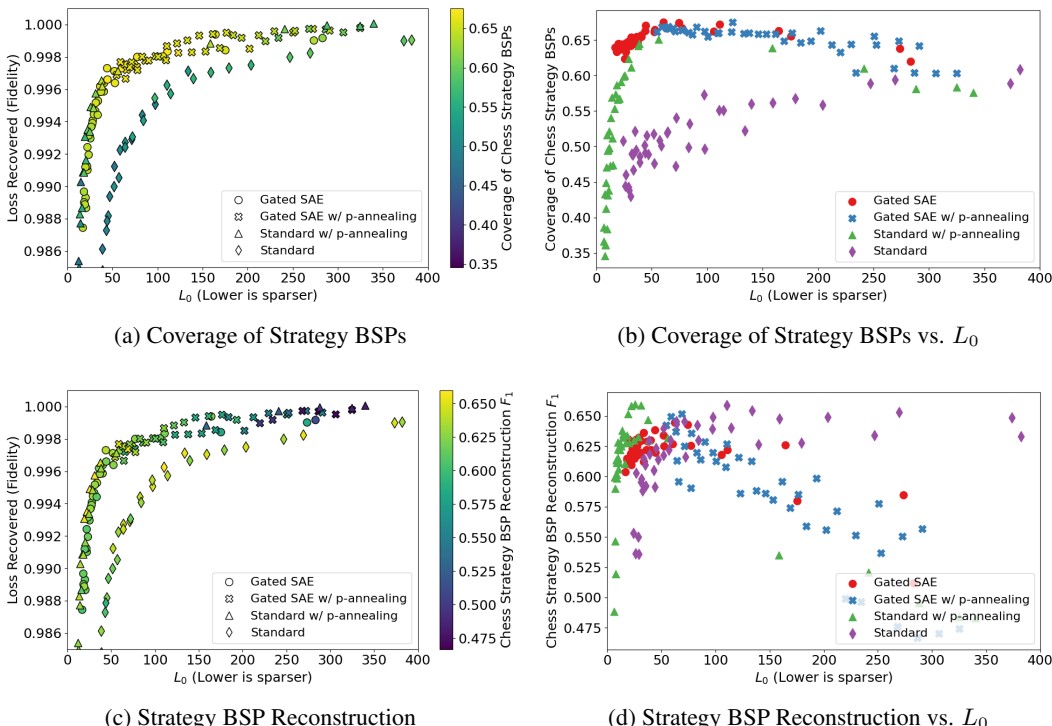

(a) Coverage of Strategy BSPs

(b) Coverage of Strategy BSPs vs. $L_0$

(c) Strategy BSP Reconstruction

(d) Strategy BSP Reconstruction vs. $L_0$

Figure 3: Comparison of the coverage and board reconstruction metrics for chess SAE quality on $\mathcal{G}_{\text{strategy}}$. The metrics represent the average coverage and board reconstruction obtained across all BSPs in $\mathcal{G}_{\text{strategy}}$. The coverage score reports the mean of maximal F1 scores over BSPs. The absolute coverage scores vary significantly between strategy BSPs, as discussed in Appendix D. The top row corresponds to coverage, and the bottom row corresponds to board reconstruction. The left column contains a scatterplot of loss recovered vs. $L_0$, with the color scheme corresponding to the coverage score and each point representing different hyperparameters. We differentiate between SAE training methods with shapes.

the larger $p$'s at the start of training, making the process worse than simply training with a constant $p$ from the beginning. Consequently, we aim to adapt the coefficient $\lambda$ such that the strength of the sparsity penalty is not changed significantly due to $p$ updates. Formally, the update step is:

$$\lambda_{s+1} \leftarrow \lambda_s \frac{\sum_{j=s-q+1}^{s} \sum_i f_i(\mathbf{x_j})^{p_s}}{\sum_{j=s-q+1}^{s} \sum_i f_i(\mathbf{x_j})^{p_{s+1}}}. \tag{14}$$

We keep a queue of the most recent $q$ batches of feature activations mid-training and use them to calibrate the $\lambda_s$ updates. Therefore, the strength of the sparsity penalty is kept *locally* constant.

**Combining $p$-annealing with other SAEs.** Since the $p$-annealing method only modifies the $L_1$ terms in the loss function without affecting the SAE architecture, it is simple to combine $p$-annealing with other SAE modifications. This allows us to create the Gated-Annealed SAE method by combining the Gated SAE architecture and $p$-annealing. Concretely, we modify $\mathcal{L}_{\text{gated}}$ (Equation 12) by replacing the sparsity term $\lambda \|\text{ReLU}(\pi_{\text{gate}}(\mathbf{x}))\|_1$ in with $\lambda_s \|\text{ReLU}(\pi_{\text{gate}}(\mathbf{x}))\|_{p_s}^{p_s}$. Our experiments showed that the optimum values for coefficients $\lambda$ and $\lambda_s$ differ.

## 5 Results

In this section, we explore the performance of SAEs applied to language models trained on Othello and chess. Consistent with Nanda et al. [47], we find that interpretable SAE features typically track properties relative to the player whose turn it is (e.g. "my king is pinned" rather than "the white king

| | Chess | | Othello | |
|---|---|---|---|---|
| **Model** | **Coverage** | **Reconstruction** | **Coverage** | **Reconstruction** |
| SAE: random GPT | 0.11 | 0.01 | 0.27 | 0.08 |
| SAE: trained GPT | 0.48 | 0.85 | 0.52 | 0.95 |
| Linear probe | 0.98 | 0.98 | 0.99 | 0.99 |

Table 1: Best performance obtained for different techniques across games for $\mathcal{G}_{\text{board state}}$. As a baseline, we train an SAE on random GPT, a version of the trained GPT model with randomly initialized weights. All models were trained on activations after the post-MLP residual connection in layer 6.

is pinned"). To side-step subtleties arising from this, we only extract our activations from the token immediately preceding white's move. Specifically, we consider SAEs trained on the residual stream activations after the sixth layer using the four methods from Section 4 (see Table 2 for additional hyperparameters). In addition to our metrics introduced above, we also make use of unsupervised metrics previously appearing in the literature [9, 16, 54]:

- $L_0$ measures the average number of active SAE active features (i.e., positive activation) on a given input.

- **Loss recovered** measures the change in model performance when replacing activations with the corresponding SAE reconstruction during a forward pass. This metric is quantified as $(H_* - H_0)/(H_{\text{orig}} - H_0)$, where $H_{\text{orig}}$ is the cross-entropy loss of the board game model for next-token prediction, $H_*$ is the cross-entropy loss after substituting the model activation $\mathbf{x}$ with its SAE reconstruction during the forward pass, and $H_0$ is the cross-entropy loss when zero-ablating $\mathbf{x}$.

Our key takeaways are as follows.

**SAE features can accurately reconstruct game boards.**  In general, we find that SAE features are effective at capturing board state information in both Othello and chess (see Table 1, Figure 2d and 4d). In contrast, SAEs trained on a model with random weights perform very poorly according to our metrics, showing that SAE performance is driven by identifying structure in the models' learned representation of game boards. Nonetheless, SAEs do not match the performance of linear probes in terms of reconstructing the board state. This performance gap suggests that SAEs do not capture all of the information encoded in the model's internal representations.

**Standard SAEs trained with $p$-annealing perform on par with Gated SAEs.**  We find that standard SAEs trained using $p$-annealing consistently perform better than those trained with a constant $L_1$ penalty (Equation 8), as measured by existing proxy metrics and in terms of improvement in coverage (see Figure 2a and 4a). In fact, standard SAEs trained using $p$-annealing show a coverage score that is comparable to Gated SAEs trained without $p$-annealing. Further, we find that both $p$-annealing and Gated SAEs significantly outperform Standard SAEs in addressing the shrinkage problem [62], as detailed in Appendix E. However, we find cases where our coverage metric disagrees with existing metrics. In Figure 2, for example, Gated SAEs perform achieve a higher *loss recovered* score than Standard SAEs trained using $p$-annealing. We emphasize that the training and inference of Gated SAEs is more computationally expensive, requiring 50% more compute per forward pass compared to Standard SAEs [54].

**Coverage and board reconstruction reveal differences in SAE quality not captured by unsupervised metrics.**  Our metrics reveal improvements in SAE performance that traditional proxy metrics fail to capture. For example, we trained SAEs with hidden dimensions 4096 and 8192 (expansion factors of 8 and 16, respectively). We expect the SAEs with 8192 hidden dimensions to perform better since they have greater capacity. However, we observe that they perform equally well according to prior unsupervised metrics (see Figures 2 a, c and 4 a, c). In contrast, our metrics reveal that SAEs with larger hidden dimensions are better. For the Standard architecture, this is reflected by the parallel lines (of purple diamonds) in Figures 2 b, d and 4 b, d. Thus, our metrics are able to capture improvements from larger expansion factors. In addition, we find that the performance of $p$-annealing closely resembles that of Gated SAEs when evaluated using standard proxy metrics; it demonstrates clear improvements under our proposed metrics.

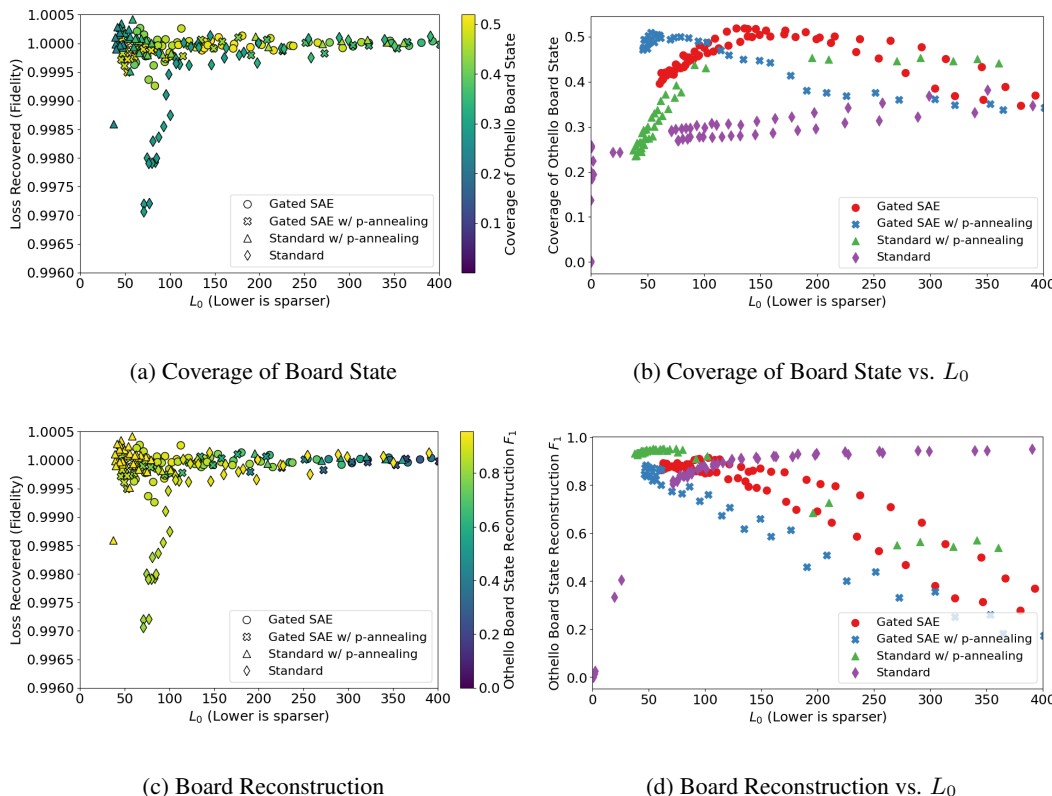

(a) Coverage of Board State

(b) Coverage of Board State vs. $L_0$

(c) Board Reconstruction

(d) Board Reconstruction vs. $L_0$

Figure 4: Comparison of the coverage and board reconstruction metrics for Othello SAE quality on $\mathcal{G}_{\text{board state}}$. The coverage score reports the mean of maximal F1 scores over BSPs. The top row corresponds to coverage, and the bottom row corresponds to board reconstruction. The left column contains a scatterplot of loss recovered vs. $L_0$, with the color scheme corresponding to the coverage score and each point representing different hyperparameters. We differentiate between SAE training methods with shapes.

**Coverage and board reconstruction are consistent with existing metrics.** Figures 2, 4, and 3 demonstrate that both coverage and board reconstruction metrics are optimal in the elbow region of the Pareto frontier. This region, where SAEs reconstruct internal activations efficiently with minimal features, also yielded the most coherent interpretations during our manual inspections. This provides precise, empirical validation to the common wisdom that SAEs in this region of the Pareto frontier are the best.

## 6   Limitations

The proposed metrics for board reconstruction and coverage provide a more objective evaluation of SAE quality than previous subjective methods. Nevertheless, these metrics exhibit several limitations. Primarily, their applicability is confined to the chess and Othello domains, raising concerns about their generalizability to other domains or different models. Additionally, the set of BSPs that underpin these metrics is determined by researchers based on their domain knowledge. This approach may not encompass all pertinent features or strategic concepts, thus potentially overlooking essential aspects of model evaluation. Developing comparable objective metrics for other domains, such as natural language processing, remains a significant challenge. Moreover, our current focus is on evaluating the quality of SAEs in terms of their ability to capture internal representations of the model. However, this does not directly address how these learned features could be utilized for downstream interpretability tasks.

# 7 Related work

**Sparse dictionary learning.** Since the nineties, dictionary learning [22, 20], sparse regression [26], and later, sparse autoencoders [49] have been extensively studied in the machine learning and signal processing literature. The seminal work of Olshausen and Field [51] introduced the concept of sparse coding in neuroscience (see also [52], building upon the earlier concept of sparse representations [19] and matching pursuit [42]. Subsequently, a series of works established the theoretical and algorithmic foundations of sparse dictionary learning [23, 30, 21, 1, 65, 32, 59, 2, 5, 7, 10]. Notably, Gregor and LeCun [28] introduced LISTA, an unrolled version of ISTA [17] that learns the dictionary instead of having it fixed.

In parallel, autoencoders were introduced in machine learning to automatically learn data features and perform dimensionality reduction [29, 37]. Inspired by sparse dictionary learning, sparse autoencoders [49, 14, 13, 41, 35] were proposed as an unsupervised learning model to build deep sparse hierarchical models of data, assuming a certain degree of sparsity in the hidden layer activations. Later, Luo et al. [39] generalized sparse autoencoders (SAEs) to convolutional SAEs. Although the theory for SAEs is less developed than that of dictionary learning with a fixed dictionary, some progress has been made in quantifying whether autoencoders can, indeed, do sparse coding, e.g., Arpit et al. [4], Rangamani et al. [55], Nguyen et al. [50].

**Feature disentanglement using sparse autoencoders.** The individual computational units of neural networks are often polysemantic, i.e., they respond to multiple seemingly unrelated inputs [3]. Elhage et al. [24] investigated this phenomenon and suggested that neural networks represent features in linear superposition, which allows them to represent more features than they have dimensions. Thus, in an internal representation of dimension $n$, a model can encode $m \gg n$ concepts as linear directions [53], such that only a sparse subset of concepts are active across all inputs – a concept deeply related to the coherence of vectors [26] and to frame theory in general [12]. To identify these concepts, Sharkey et al. [57] used SAEs to perform dictionary learning on a one-layer transformer, identifying a large (overcomplete) basis of features. Cunningham et al. [16] applied SAEs to language models and demonstrated that dictionary features can be used to localize and edit model behavior. Marks et al. [44] proposed a scalable method to discover sparse feature circuits, as opposed to circuits consisting of polysemantic model components, and demonstrated that a human could change the generalization of a classifier by editing its feature circuit. Recently, Kissane et al. [34] explored autoencoders for attention layer outputs. These works have benefited from a variety of open-source libraries for training SAEs for LLM interpretability [43, 8, 15].

# 8 Conclusion

Most SAE research has relied on proxy metrics such as loss recovered and $L_0$, or subjective manual evaluation of interpretability by examining top activations. However, proxy measures only serve as an estimate of interpretability, monosemantic nature, and comprehensiveness of the learned features, while manual evaluations depend on the researcher's domain knowledge and tend to be inconsistent.

Our work provides a new, more objective paradigm for evaluating the quality of an SAE methodology; coverage serves as a quantifiable measure of monosemanticity and quality of feature extraction, while board reconstruction serves as a quantifiable measure of the extent to which an SAE is exhaustively representing the information contained within the language model. Therefore, the optimal SAE methodology can be judged by whether it yields both high coverage and high board reconstruction.

Finally, we propose the $p$-annealing method, a modification to the SAE training paradigm that can be combined with other SAE methodologies and results in an improvement in both coverage and board reconstruction over the Standard SAE architecture.

## Author Contributions

A.K. built and maintained our infrastructure for working with board-game models. A.K., S.M., C.R., J.B., and L.S. designed the proposed metrics. B.W. performed initial experiments demonstrating the benefits of training SAEs with $p < 1$. B.W., C.M.V., and S.M. then proposed $p$-annealing, with B.W. leading the implementation and developing coefficient annealing. The basic framework for our dictionary learning work was built and maintained by S.M. and C.R. The training algorithms studied were implemented by S.M., C.R., B.W., R.A., and J.B. R.A. trained the SAEs used in our experiments. A.K., C.R., and J.B. selected and implemented the BSPs. A.K. and J.B. trained the linear probes. Many of the authors (including L.S., J.B., R.A.) did experiments applying traditional dictionary learning methods and exploring both toy problems and natural language settings, which helped build valuable intuition. The manuscript was primarily drafted by A.K., B.W., C.R., R.A., J.B., C.M.V., and S.M., with extensive feedback and editing from all authors. D.B. suggested the original project idea.

## Acknowledgments

C.R. is supported by Manifund Regrants and AISST. L.R. is supported by the Long Term Future Fund. S.M. is supported by an Open Philanthropy alignment grant.

The work reported here was performed in part by the University of Massachusetts Amherst Center for Data Science and the Center for Intelligent Information Retrieval, and in part using high performance computing equipment obtained under a grant from the Collaborative R&D Fund managed by the Massachusetts Technology Collaborative.

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

# A  Improving and evaluating sparse autoencoders

Despite the success of SAEs at extracting human-interpretable features, they fail to perfectly reconstruct the activations [16]. One challenge in the training of SAEs with an $L_1$ penalty is shrinkage (or 'feature suppression'); in addition to encouraging sparsity, an $L_1$ penalty encourages feature activations to be smaller than they would be otherwise. Wright and Sharkey [62] approached this problem by fine-tuning the sparse autoencoder without a sparsity penalty. Appendix E further quantifies shrinkage across a suite of SAEs trained on chess and Othello models. Jermyn et al. [31] and Riggs and Brinkmann [56] explored alternative sparsity penalties to reduce feature suppression during training. Rajamanoharan et al. [54] introduced Gated SAEs, an architectural variation for the encoder which both addresses shrinkage and improves on the Pareto frontier of $L_0$ vs reconstruction error. Recently, Gao et al. [27] systematically evaluated the scaling laws with respect to sparsity, autoencoder size, and language model size.

The goal of dictionary learning in machine learning is to produce human-interpretable features and capture the underlying model's computations [9]. However, quantitatively measuring interpretability is difficult and often involves manual inspection. Therefore, most existing work assesses the quality of SAEs along different proxy metrics: (1) The cross-entropy loss recovered, which reflects the degree to which the original loss of the language model can be recovered when replacing activations with the autoencoder predictions. (2) The $L_0$-norm of feature activations $\mathbb{E}_{z \sim \mathcal{D}} \|h(z)\|_0$, measuring the number of activate features given an input [25]. Makelov et al. [40] proposed to compare SAEs against supervised feature dictionaries in a natural language setting. However, this requires a significant understanding of the model's internal computations and is thus not scalable.

# B  Sparse Autoencoder Training Parameters

We used a single NVIDIA A100 GPU for training SAEs and experiments. It takes much less than 24 hours to train a single SAE on 300 million tokens. Given a trained SAE, our evaluation requires less than 5 minutes of computing time.

Table 2: Training parameters of our sparse autoencoders.

| Parameter | Value |
|---|---|
| Number of tokens | 300M |
| Optimizer | Adam |
| Adam betas | (0.9, 0.999) |
| Linear warmup steps | 1,000 |
| Batch size | 8,192 |
| Learning rate | 3e-4 |
| Expansion factor | {8, 16} |
| Annealing start | 10,000 |
| $p_{\text{end}}$ | 0.2 |
| $\lambda_{\text{init}}$ | [0.02, 2.0] |

# C   List of Board State Properties

Table 3 summarizes the high-level board state properties considered in $\mathcal{G}_{\text{strategy}}$. The selection of concepts was inspired by McGrath et al. [45]. The column indicated by # denotes the number of individual BSPs per concept. A single BSP per concept indicates we match this condition globally for any corresponding piece.

Table 3: List of strategic Board State Properties.

| Concept | # | Description |
|---|---|---|
| check | 1 | Indicates whether the player to move is checked by the opponent. |
| can_check | 1 | Indicates whether the player to move could check the opponent with the next move. |
| queen | 1 | Indicates whether the player to move has a queen on the board. |
| can_capture_queen | 1 | Indicates whether the player to move can capture the queen of the opponent. |
| bishop_pair | 1 | Indicates whether the player to move still has both bishops on the board. |
| castling_rights | 1 | Indicates whether the player to move is still allowed to castle, contingent on the king and the rooks not having moved. |
| kingside_castling_rights | 1 | Indicates whether the player to move is still allowed to kingside castle, contingent on the king and the kingside rook not having moved. |
| queenside_castling_rights | 1 | Indicates whether the player to move is still allowed to queenside castle, contingent on the king and the queenside rook not having moved. |
| fork | 1 | Indicates whether the player to move attacks has a fork on major pieces of the opponent. |
| pin | 1 | Indicates whether there is a pin on the board, such that a player's piece cannot move without exposing the king behind it to capture. |
| legal_en_passant | 1 | Indicates whether the player to move has a legal en passant: a special pawn capture that can only occur immediately after an opponent moves a pawn two squares from its starting position and it lands beside the player's pawn. |
| ambiguous_moves | 1 | Indicates whether there are moves that would require further specification as more than one piece of the same type can move to the same square. |
| threatened_squares | 64 | Indicates which squares are threatened by the opponent. |
| legal_moves | 64 | Indicates which squares can be legally moved to by the current player. |

# D Performance of Linear Probes and SAEs on Board State Properties

In Figure 3, we present a mean coverage score over strategy board state properties $\mathcal{G}_{\text{BSP}}$. Properties within $\mathcal{G}_{\text{BSP}}$ vary significantly in complexity. For example, queen detection can be inferred directly from the move history, while fork detection requires an accurate representation of the board state. Table 4 shows that linear probe F1-score is below 0.95 for 6 out of 15 properties in $\mathcal{G}_{\text{BSP}}$. This suggests that chess-GPT [33] does not represent these properties linearly. Additional experiments are required to determine whether the representation is present at all.

For the board state case, reconstruction is significantly higher than coverage. This is because there are many SAE features that are high precision classifiers for a configuration of squares, such as "white pawn on `e4`, white knight of `f3`". In cases where coverage is higher than reconstruction (such as for `can_check`), it is because there are not many features that are over 95% precision for "there is a *check move* available" from which we can recover if there is an available check move. Coverage is significantly higher because there is at least one feature that has an $F_1$-score of 0.54 for `can_check`, which may not have a precision greater than 95%.

Table 4: Comparison of performance of linear probes trained to predict board state properties given residual stream activation of ChessGPT after the sixth layer with SAEs evaluated using our coverage and reconstruction metrics.

| Concept | Linear Probe $F_1$-score | Best SAE Reconstruction score | Best SAE Coverage score |
|---|---|---|---|
| `check` | 1.00 | 1.00 | 1.00 |
| `can_check` | 0.93 | 0.27 | 0.54 |
| `can_capture_queen` | 0.66 | 0.62 | 0.48 |
| `queen` | 1.00 | 0.97 | 0.96 |
| `bishop_pair` | 1.00 | 0.83 | 0.86 |
| `castling_rights` | 1.00 | 0.98 | 0.82 |
| `kingside_castling` | 1.00 | 0.98 | 0.81 |
| `queenside_castling` | 1.00 | 0.97 | 0.81 |
| `fork` | 0.68 | 0.13 | 0.38 |
| `pin` | 0.67 | 0.20 | 0.33 |
| `legal_en_passant` | 0.96 | 0.92 | 0.90 |
| `ambiguous_moves` | 0.72 | 0.25 | 0.57 |
| `threatened_squares` | 0.96 | 0.93 | 0.71 |
| `legal_moves` | 0.92 | 0.66 | 0.63 |
| `board_state` | 0.98 | 0.67 | 0.41 |

Table 5: Comparison of performance of linear probes trained to predict high-level board state properties given residual stream activations with SAEs, both trained on a model with the same architecture as ChessGPT but randomly initialized. Performance on metrics can be high when the metric is correlated with move number or syntax level patterns (such as castling, which corresponds to "0-0").

| Concept | Linear Probe $F_1$-score | Best SAE Reconstruction score | Best SAE Coverage score |
|---|---|---|---|
| check | 0.00 | 0.00 | 0.13 |
| can_check | 0.19 | 0.03 | 0.52 |
| can_capture_queen | 0.00 | 0.00 | 0.09 |
| queen | 0.85 | 0.95 | 0.93 |
| bishop_pair | 0.82 | 0.74 | 0.81 |
| castling_rights | 0.89 | 0.75 | 0.65 |
| kingside_castling | 0.89 | 0.75 | 0.65 |
| queenside_castling | 0.89 | 0.75 | 0.64 |
| fork | 0.01 | 0.00 | 0.07 |
| pin | 0.00 | 0.00 | 0.25 |
| legal_en_passant | 0.00 | 0.00 | 0.06 |
| ambiguous_moves | 0.13 | 0.00 | 0.52 |
| threatened_squares | 0.82 | 0.73 | 0.60 |
| legal_moves | 0.65 | 0.36 | 0.45 |
| board_state | 0.26 | 0.01 | 0.11 |

# E   Relative Reconstruction Bias

Training Standard SAEs with an $L_1$ penalty, as described in Section 4, causes a systematic underestimation of feature activations. Wright and Sharkey [62] term this phenomenon *shrinkage*. Following Rajamanoharan et al. [54], we measure the relative reconstruction bias $\gamma$ of our SAEs, defined as:

$$\gamma := \arg\min_{\gamma'} \mathbb{E}_{x \sim \mathcal{D}} \left[ \| \hat{x}_{\text{SAE}}(x)/\gamma' - x \|_2^2 \right] \tag{15}$$

Here, $\mathcal{D}$ denotes a large dataset of model internal activations. Intuitively, $\gamma < 1$ indicates shrinkage. A perfectly unbiased SAE would have $\gamma = 1$.

Our experiments show that p-annealing achieves similar relative reconstruction bias improvements to gated SAEs, both outperforming the baseline architecture. Figure 5 shows that improvements manifest differently across domains: Chess SAEs show a narrower range of bias ($\gamma \approx 0.98$) compared to Othello ($\gamma \approx 0.80$). This domain-dependent variation may reflect differences in the underlying models or data distributions. We observe unstable $\gamma$ values for SAEs with L0 near zero, which represent degenerate cases outside the typical operating range of these models.

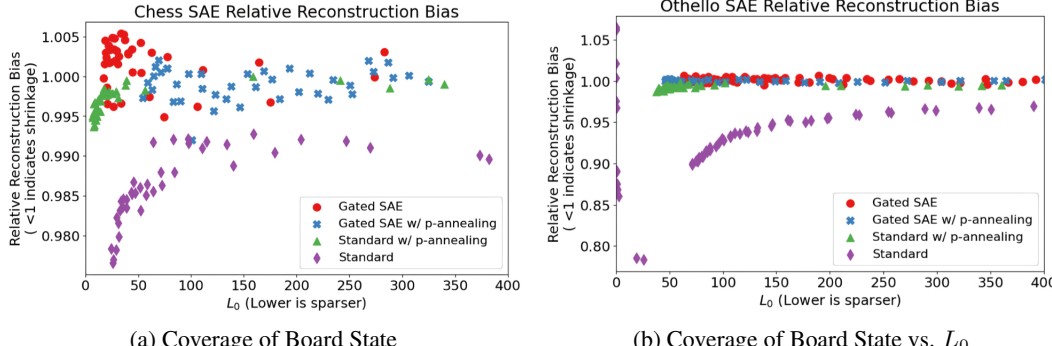

(a) Coverage of Board State    (b) Coverage of Board State vs. $L_0$

Figure 5: Comparison of the relative reconstruction bias metric $\gamma$ quantifying feature activation shrinkage across a suite of SAEs. $\gamma < 1$ indicates shrinkage. A perfectly unbiased SAE would have $\gamma = 1$.

# F    Model Internal Board State Representation

## F.1    Othello Models

Previous research of Othello-playing language models found that the model learned a nonlinear model of the board state [36]. Further investigation found a closely related linear representation of the board when probing for "my color" vs. "opponent's color" rather than white vs. black [47]. Based on these findings, when measuring the state of the board in Othello, we represent squares as (`mine`, `yours`) rather than (`white`, `black`).

## F.2    Chess Models

Similar to Othello models, prior studies of chess-playing language models found the same property, where linear probes were only successful on the objective of the (`mine`, `yours`) representation and were unsuccessful on the (`white`, `black`) representation [33]. They measured board state at the location of every period in the Portable Game Notation (PGN) string, which indicates that it is white's turn to move and maintain the (`mine`, `yours`) objective. Some characters in the PGN string contain little board state information as measured by linear probes, and there is not a clear ground truth board state part way through a move (e.g., the "`f`" in "`Nf3`"). We follow these findings and measure the board state at every period in the PGN string.

When measuring chess piece locations, we do not measure pieces on their initial starting location, as this correlates with position in the PGN string. An SAE trained on residual stream activations after the first layer of the chess model (which contains very little board state information as measured by linear probes) obtains a board reconstruction $F_1$-score of 0.01 in this setting. If we also measure pieces on their initial starting location, the layer 1 SAE's $F_1$-score increases to 0.52, as the board can be mostly reconstructed in early game positions purely from the token's location in the PGN string. Masking the initial board state and blank squares decreases the $F_1$-score of the linear probe from 0.99 to 0.98.

# G    Additional examples of learned SAE features

We present two additional examples of learned SAE features that we (subjectively) match to board state properties based on their maximally activating input PGN-strings in Figure 6.

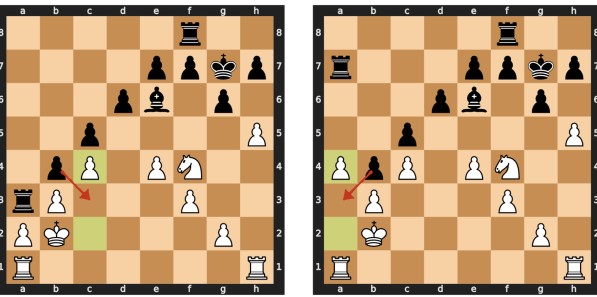

```
1.e4 c5 2.Nc3 Nc6 3.Nf3 g6 4.d4 cxd4 5.Nxd4 Bg7 6.Be3 Nf6 7.Qd2 Ng4
8.Nxc6 bxc6 9.Bd4 Bxd4 10.Qxd4 0-0 11.Be2 d6 12.Bxg4 Bxg4 13.f3 Be6
14.h4 Qb6 15.0-0-0 Rab8 16.Qxb6 axb6 17.h5 Kg7 18.b3 b5 19.Kb2 b4
20.Ne2 c5 21.Nf4 Ra8 22.Ra1 Ra3 23.c4 Ra7 24.a4
```

(a) "En passant available" detector learned by an SAE.

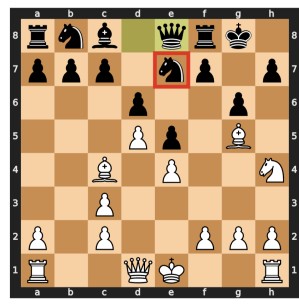

```
1.e4 e6 2.Nf3 d6 3.Nc3 Ne7 4.d4 g6 5.Bg5 Bg7
6.Bc4 0-0 7.d5 Bxc3+ 8.bxc3 e5 9.Nh4 Qe8 10.Q
```

(b) "Knight on e2 or e7" detector learned by
an SAE.

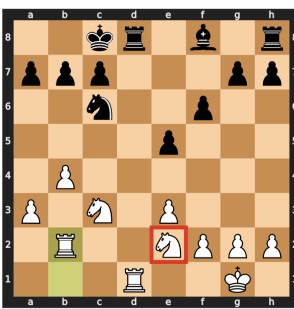

```
1.d4 d5 2.c4 Nc6 3.cxd5 Qxd5 4.Nc3 Qxd4 5.e3 Qxd1+ 6.Nxd1 Bg4 7.Be2 Bxe2
8.Nxe2 0-0-0 9.0-0 e5 10.a3 Nf6 11.b4 Ne4 12.Bb2 f6 13.Ndc3 Nd2 14.Rfd1 Nc4
15.Rab1 Nxb2 16.Rxb2
```

(c) Another example of the "Knight on e2 or e7" detector shown
in subfigure (b) above. We interpret this feature as representing a
mirrored perspective. It may be firing for "opponent knight on e
column, 1 square away from opponent back rank", which is why it
fires for both e2 (if black's turn to move) and e7 (if white's turn to
move).

Figure 6: Additional examples of learned SAE features. We show the full board state of a chosen
game in which the SAE latent has a high activation. The PGN-string (model input) which represents
the game history is shown below the board. Tokens that activate SAE features are marked in blue,
where darker shades correspond to higher feature activations. Moves that create the considered a
board state are highlighted in yellow.

