# OpenReview forum: "Measuring Progress in Dictionary Learning for Language Model Interpretability with Board Game Models"
_NeurIPS.cc/2024/Conference — NeurIPS 2024 poster_

### Official Review · Reviewer_pBfq · 2024-07-09

**Soundness:** 2
**Presentation:** 2
**Contribution:** 3
**Rating:** 5
**Confidence:** 4

**Summary:**

This paper proposes two measures to reflect the quality of the trained sparse auto-encoder when decomposing superposition features. In addition, this paper achieves approximate L0 optimization through dynamic adjustment of the p-norm and testing it on two types of board games.

**Strengths:**

The research direction of this article is very valuable. Previous work using SAE for feature detection and circuit discovery did not directly measure the interpretability of the model decomposition results but relied on indirect measures such as reconstruction and sparsity levels of SAE. This work proposes using a few rules (low or high level) to automatically analyze the results of SAE decomposition under different hyperparameters, which is helpful for subsequent selection of SAE used for feature detection. Additionally, dynamically adjusting the p-norm is a straightforward method that intuitively approximates optimization of L0.

**Weaknesses:**

- The experimental objectives of the article are not clear: The article lacks of comparing the advantages of the two proposed measurement methods compared to reconstruction loss and sparsity. For example, demonstrating cases where both reconstruction loss and sparsity perform well but coverage and reconstruction are low, resulting in biases in downstream tasks (such as circuit discovery).
- The rationality of the two proposed measurement methods in the article is questionable: The purpose of using SAE is to decompose the features of superposition into monosemantic features, with the expected decomposition results leaning towards several low-level features. However, based on the experimental results in Figure 2, there is no significant difference between low-level and high-level coverage results. This phenomenon is likely due to the unreasonable selection of rules.
- Similarly, due to the fact that both coverage and reconstruction are ultimately detected using linear probing on individual features, some high-level features require linear combinations of multiple low-level features to be expressed [1,2,3]. If the measurement methods described in the article are used, it will not be effective in detecting these features.
- Typos (not limited to): Line 154  "than than" should be corrected.
- The image captions are incorrect: Figure 2 should be labeled "coverage" instead of "reconstruction" for the bottom row.


[1] Templeton, Adly, et al. "Scaling monosemanticity: Extracting interpretable features from claude 3 sonnet." Transformer Circuits Thread (2024).

[2] He, Zhengfu, et al. "Dictionary Learning Improves Patch-Free Circuit Discovery in Mechanistic Interpretability: A Case Study on Othello-GPT." arXiv preprint arXiv:2402.12201 (2024).

[3] Rajamanoharan, Senthooran, et al. "Improving dictionary learning with gated sparse autoencoders." arXiv preprint arXiv:2404.16014 (2024).

**Questions:**

- Is there a separate training of SAE for MLPs, Attention outputs, etc., which write into the residual stream instead of activations between attention blocks?
- Is there any relevant reference for the assumption about *High Precision* in line 130?
- How were the hyperparameters adjusted in Figures 2, 3, 4, and 5 to obtain several different SAEs? (Causality between hyperparameters and SAE results needs further clarification [1].)
- In [2], it's mentioned that Gated SAE can solve the shrinkage problem mentioned in line 167. Have you compared the relative reconstruction bias (gamma) proposed in [2] when using gated SAE w/wo p-annealing?

[1] Kramár, János, et al. "AtP*: An efficient and scalable method for localizing LLM behaviour to components." arXiv preprint arXiv:2403.00745 (2024).

[2] Rajamanoharan, Senthooran, et al. "Improving dictionary learning with gated sparse autoencoders." arXiv preprint arXiv:2404.16014 (2024).

**Limitations:**

I suggest the authors expand the scope of the research to include more domains and datasets, and validate the effectiveness of the proposed methods in other downstream interpretability tasks.

---

> ### Author Rebuttal · Authors · 2024-08-07
>
> Thank you for your thoughtful review. We’re glad that you think the research direction in our paper is very valuable.
>
> **Comparison of the advantages of the two proposed measurement methods compared to reconstruction loss and sparsity:** Thank you for giving us the opportunity to clarify this. In our paper, we compare three SAE training methodologies: Gated, Gated with p-annealing, and Standard with p-annealing. All three show Pareto improvements over the Standard approach in terms of reconstruction loss and sparsity but cannot be distinguished using existing unsupervised metrics (L0 and fraction of loss recovered). However, as you can see in the pdf attached to our global comment, the metrics we introduce clearly distinguish the SAEs. In other words, our metrics reveal differences in SAE quality which are invisible to prior unsupervised metrics. (As an especially clear example, the parallel curves that appear in our plots correspond to SAEs trained with the same algorithm but different hidden dimensions; our metrics clearly demonstrate that the SAEs with larger hidden dimensions are better than those with smaller hidden dimensions, as expected.)
>
> **No significant difference between low-level and high-level coverage, unreasonable selection of rules:** Thank you for pointing this out. In our original submission there was a mismatch between graphs and captions, and our actual high-level coverage graph was not included. We have provided an updated selection of graphs in the PDF.
>
> We also agree that our initial split into low vs. high seemed ambiguous and subjective. To address this, we moved to a more principled division of BSPs into (1) board state BSPs, which correspond to the presence of a particular piece at a particular board space (giving 8 x 8 x 12 board state BSPs for chess and 8 x 8 x 2 for Othello), and (2) researcher-selected strategy BSPs for chess. Our goal was to confine researcher choice to the strategy BSPs as much as possible, while keeping the class of board state BSPs natural and principled.
>
> **Some high-level features require linear combinations of multiple low-level features to be expressed:** We would like to clarify that linear probing is not used in either the coverage or reconstruction metrics, which instead measure whether individual features found using SAEs correspond to individual BSPs. From our perspective, neither our work nor prior work suggests that interpretable high-level properties are necessarily expressed using an ensemble of low-level features. For example, in our work, we find individual features corresponding to high-level properties such as “an en passant capture is available”. Similarly, in Anthropic’s Scaling Monosemanticity [4] they find “a diversity of highly abstract features”. We hope this addresses your concern, but we also acknowledge that we might be misinterpreting your question. If this is the case, could you please provide further clarification?
>
> **Training of SAEs on various locations:** We only train SAEs on the residual stream between layers 6 and 7, and do not train SAEs on other locations. We chose this location because it achieved the best performance when reconstructing the board state with linear probes, which is consistent with prior work [1, 2, 3].
>
> **Assumption of High Precision:** We had several motivations for choosing high precision features. The first was that during manual inspection of SAE features, we found that features often seemed to activate on specific board configurations, rather than individual board square states. We also noted studies on chess players showing that chess experts excel at remembering realistic board configurations, but not random piece placements[4]. This suggests experts (and potentially AI models) encode board states as meaningful patterns rather than individual square occupancies.
>
> Concurrently, Anthropic has also evaluated features on their precision in Scaling Monosemanticity, which they called “specificity” [5].
>
> **Hyperparameter adjustment:** We sweep over the hyperparameters of sparsity penalty, learning rate, and expansion factor. A lower sparsity penalty tends to increase L0, while a larger expansion factor tends to increase Loss Recovered.
>
> **Relative Reconstruction Bias (Gamma from GDM paper [6])** Thank you for this suggestion. We have performed additional experiments measuring relative reconstruction bias. Our results show p-annealing achieves similar relative reconstruction bias to gated and gated w/ p-annealing, all improving over the standard SAE. The results can be seen in Figure 4 of our global pdf. We note that the y-axis scale for relative reconstruction bias differs between Chess and Othello SAEs. Chess shows a narrower range (minimum γ ≈ 0.98) compared to Othello (minimum γ ≈ 0.80), which more closely resembles the original GDM plot [6]. This may reflect differences in the underlying models or data.  In the Othello plot, we found erratic relative reconstruction bias values for SAEs with L0s very near 0; these are very bad SAEs outside of the sparsity range usually considered (including in [6]).
>
> [1] K. Li, A. K. Hopkins, D. Bau, F. Viégas, H. Pfister, and M. Wattenberg, ‘Emergent World Representations: Exploring a Sequence Model Trained on a Synthetic Task’, arXiv [cs.LG]. 2024.
>
> [2] N. Nanda, A. Lee, and M. Wattenberg, ‘Emergent Linear Representations in World Models of Self-Supervised Sequence Models’, arXiv [cs.LG]. 2023.
>
> [3] A. Karvonen, ‘Emergent World Models and Latent Variable Estimation in Chess-Playing Language Models’, arXiv [cs.LG]. 2024.
>
> [4] Frey PW, Adesman P. ‘Recall memory for visually presented chess positions.’ Mem Cognit. 1976.
>
> [5] A. Templeton et al., ‘Scaling Monosemanticity: Extracting Interpretable Features from Claude 3 Sonnet’, Transformer Circuits Thread, 2024.
>
> [6] Rajamanoharan, Senthooran, et al. "Improving dictionary learning with gated sparse autoencoders." arXiv preprint arXiv:2404.16014 (2024).

---

> > ### Comment · Reviewer_pBfq · 2024-08-13
> > **Reply for Authors**
> >
> > I appreciate the authors' response. My main question focuses on the first point. Let me briefly explain my understanding of the first question.
> > > The author believes that the existing L0 and reconstruction loss cannot accurately measure the training results of SAE (**I also agree**), Therefore, the author proposed two new measurement methods and observed that the extreme value area of ​​the new measurement method is at the Pareto front of the original measurement method, so the two new measures are considered to be better than the original method.
> >
> > But this only shows that the new measurement method considers L0 and reconstruction loss at the same time (similar to harmonic averaging), but does not reflect the loopholes of the original measurement (for example, different training settings obtained two groups of L0 and reconstruction loss with the same SAE, but the subsequent performance of these two groups of SAE is very different). And this is also my main concern with this article. Therefore, I maintain my rating.
> > And I am open to revising my rating based on discussions with other reviewers/ AC.

---

> > > ### Author Response · Authors · 2024-08-14
> > >
> > > The reviewer summarizes our original response as
> > >
> > > > The author believes that the existing L0 and reconstruction loss cannot accurately measure the training results of SAE (I also agree), Therefore, the author proposed two new measurement methods and observed that the extreme value area of ​​the new measurement method is at the Pareto front of the original measurement method, so the two new measures are considered to be better than the original method.
> > >
> > > To be clear, while it is true that one of our results is that our new supervised metrics obtain their best values in the "elbow" region of the Pareto frontier for unsupervised proxy metrics (L0 and loss recovered), this is **not** the reason we claim our supervised metrics improve on prior unsupervised metrics. We include this result as a "sanity check" to show that our metrics do not totally diverge from prior notions of SAE quality. (And we agree that—as the reviewer's "harmonic averaging" example shows—this result alone leaves open the possibility that our supervised metrics contribute nothing new.)
> > >
> > > Rather, the reasons our metrics improve on prior measures of SAE quality are:
> > >
> > > 1. (Empirical result) Our metrics show differences between SAEs which are invisible to prior unsupervised metrics (see the pdf attached to our general response). As discussed in our original response, an especially striking example is given by the parallel curves that appear in our plots: our metrics clearly show that SAEs with a larger hidden dimension are better, but prior unsupervised metrics failed to show this.
> > >
> > > 2. (Theoretical property) Our metrics are supervised—based on researcher operationalization of what counts as an "interpretable" feature and which features we expect good SAEs to learn—whereas prior SAE quality measures (L0 and loss recovered) are unsupervised.

---

> > > > ### Comment · Reviewer_pBfq · 2024-08-14
> > > > **Reply for Authors**
> > > >
> > > > Thanks to the author for the reply, the two points listed answered my question, so I will increase my score.

---

### Official Review · Reviewer_R4sd · 2024-07-10

**Soundness:** 3
**Presentation:** 4
**Contribution:** 3
**Rating:** 7
**Confidence:** 4

**Summary:**

This manuscript applies sparse autoencoders (SAE) to detect interpretable features from autoregressive transformer models trained on Othello and chess. These controlled scenarios provide suitable testbeds, in the sense that we can extract ground truth features to measure progress in dictionary learning. Based on the observations that existing SAE optimizations may indeed be suboptimal, the authors propose *p-annealing*, a warm-start technique that iteratively optimize a sequence of (increasingly more non-convex) SAE objectives, in order to approximate the intractable $\ell_0$ objective.

**Strengths:**

* The presentation of this paper is very clear. I really enjoyed reading this paper overall.
* The topic of this paper is of high relevance NeurIPS community. SAEs are shown to be a promising approach for mechanistic interpretability. Providing a quick-to-iterate testbed as well as a set of improvement techniques would surely be valuable to this research.
* The $p$-annealing technique is reasonable, is easy to implement, and has rich history in sequential and/or global optimization. It also demonstrate superior performance compared to standard SAE objectives.
* Experiments are thorough, and demonstrate (1) the applicability of SAEs to learn interpretable features for Othello and chess tasks, and (2) the benefits of improved techniques on these tasks.

**Weaknesses:**

* While it is reasonable to operate in a controlled setup (i.e. Othello and chess) in this paper, there lacks concrete evidence on how these improved techniques can potentially improve SAEs in LLMs and natural language domains. Including a discussion and/or additional experiments on this could further strengthen the paper.

**Questions:**

* How sensitive is the SAE training to the choice of $p$-annealing schedule and other choice of hyperparameters?

**Limitations:**

The authors have adequately addressed limitations and potential societal impacts of their work

---

> ### Author Rebuttal · Authors · 2024-08-07
>
> Thank you for your review. We’re glad that you enjoyed reading the paper and think it’s of high relevance to the NeurIPS community.
>
> **Demonstrating transfer to natural language:** Yes, we agree that this is a limitation in our paper and future work should study this. However, note that our metrics reveal progress in SAE training which is invisible to existing proxy metrics. Specifically, SAEs with larger hidden dimensions perform better on our metrics, e.g. for the Standard architecture this is reflected in the parallel lines of purple diamonds (see attached PDF). The hope would be that training techniques which work better in the board game setting also work better in the natural language setting, though we don't validate that in this paper.
>
> **Sensitivity of SAE training to p-annealing schedule and other hyperparameters:** We did not conduct a systematic evaluation of the p-annealing schedule and other hyperparameters. From our experience it did not seem that sensitive to the schedule, but is a bit sensitive to the p_end value, i.e. the final p value at which we would stop the training. If p_end was below 0.15, the training started to destabilize.
>
> We also noticed an effect on the initial lambda coefficient (strength of the sparsity penalty). Specifically, when using p-annealing, a small change to lambda could lead to larger differences in L0 than when training without it. This means it would often tighten the range of lambdas that we would explore. However, this only meant that we would explore over less orders of magnitude of lambda coefficients and get the same L0 spread.

---

> > ### Comment · Reviewer_R4sd · 2024-08-12
> >
> > Thank you for these feedbacks. I maintain my current score and look forward to seeing future works on applying this type of SAEs to natural language setups.

---

### Official Review · Reviewer_NNJv · 2024-07-13

**Soundness:** 3
**Presentation:** 3
**Contribution:** 3
**Rating:** 7
**Confidence:** 3

**Summary:**

The paper introduces a setting for evaluating dictionary learning methods (and in particular sparse autoencoders) for language model interpretability. The setting proposed is that of interpreting features learned by language models trained on data representing Chess and Othello games. This setting should allow testing based on a more easily enumerable set of ground truth features (compared to natural language) which we know are relevant to the task and are captured by the underlying model.

The authors propose two new F1 based metrics to evaluate the SAEs ability to recover features learned by the model, namely a "coverage" metric and a "board reconstruction" metric that measures the quality of high precision features in the SAE. Both these metrics are computed by treating SAE features as classifiers for binary board state properties.

The paper also introduces a new training technique for SAEs called p-annealing that modifies the L1 loss to more closely approximate optimizing the L0 norm for which L1 loss acts as a proxy.

**Strengths:**

- Having a better grasp on how to evaluate SAE's training methods would be very useful as they are generally fairly difficult objects to evaluate in terms of how interpretable they are (without a lot of manual inspection). The setting used is complex enough to be interesting while maintaining an ability to know ground truth concepts that should be present and separately probe that the models the SAEs are trying to give insight into actually have representations of those concepts.
- The metrics proposed intuitively make sense and appear to relate well to established metrics like precision, recall and F1.
- The motivation for p-annealing is well thought out and presented. And the method appears to have a strong effect on lowering overall sparsity.

**Weaknesses:**

- The figures and caption for figure 2 don't seem to match up (there may be a mixup with subfigures for figure 3). From reading the caption I would have expected the bottom row to show "Coverage" for high level BSPs in a similar pattern to the top row, however it shows "Reconstruction" of high level BSPs and reconstruction of low-level BSPs (which seem to appear as subfigures in figure 3). Could the authors correct this/post the appropriate charts?

- I can't easily understand the comparison of the new metrics compared to the existing proxy metrics, the figures (2-5) seem to attempt to both validate the new metrics as well as compare the proposed p-annealing technique and thus get fairly busy making them hard to parse visually:
    - The 3 variable plots on the left hand column of the figures are somewhat hard to read, especially since the proposed metric is encoded using color. Having a plot comparing each proxy metric (loss recovered and sparsity) to each proposed metrics would be easier to read. I do understand that the authors are trying to demonstrate that there is a region on the Pareto frontier where their metrics perform best, its not so apparent to me that sparser SAEs are always more "interpretable" SAEs so it is an interesting finding, but I still think it would be helpful to more easily compare each metric to the existing proxy metrics (for example I wouldn't necessarily expect that less sparse SAEs should have worse coverage—but maybe the authors have some thoughts on why that should be the case?)
    - Having separate plots that focus on evaluating the p-annealing technique using the various metrics (proxy and proposed) when also be helpful (i.e. similar to Fig 5 in Rajamanoharan et al 2024). The overplotting of the various shapes for the different conditions (particularly in the left hand column) makes it a bit hard to understand the the trends and understand how well p-annealing works actually works.
- From my read of the charts, the results for p-annealing seem quite different between Othello and Chess. I'm trying to map the final sentence about the efficacy p-annealing in the conclusion to the charts and i'm not sure if i'm able to see how the conclusion follows from the results. Could the authors comment on this?

- There isn't much description of qualitative/manual analysis of the trained SAEs alongside the new metrics. Because measurement is relatively nascent in this space, it would be helpful to have some qualitative evaluation/results from manual inspection demonstrated.

**Questions:**

- What is functionally/representationally different about BSPs in F_low vs F_high? "Any square being threatened" seems as high level as "has a queen on the board". I'm just trying to understand if there is a critical difference between these two classes of BSP from the perspective of the SAE or BSP classifier. It is nice that these concepts come from different sources, just wondering about the low-level vs high level implication.
- What is the effect on the board reconstruction metric if there are BSPs for which there are no high precision features (line 131)?  Are these low precision features still used in calculating the final metric? It was not super clear to me from the notation in the equation below line 134 if **all** features are used used in this calculation, _but_ φfi,t(x) = 1 is only calculated for high precision features, or if **only** high precision features are included in this computation at all?
- Why use a "batch of train data" to identify the high precision features? Why not use the whole dataset? How big should this batch be relative to the whole dataset?
- Have the authors done any qualitative analysis of their metrics?
	- For example board reconstruction seems to overall be lower for high-level vs low-level BSPs. Have the authors observed that this translates into more difficulty interpreting board states retrieves by featured for the high level BSPs?
- How many SAEs are trained for each of the 4 SAE training conditions {Gated, Standard} x {p-annealing, no p-annealing}?
- Could you elaborate on line 220 where you say "in the region with L0 below 100, which is the optimal range according to proxy metrics"? Is this a reference to an existing finding, or how do we know that this is the optimal range for sparsity?

A more speculative question:
- Do the authors know if there is any correlation between their proposed metrics and the final reconstruction loss of the SAE? Particuarly since the linear probe performance on predicting BSPs seems to be high, might we expect to see a correlation to just straight reconstruction loss as another signal to validate these metrics or am I off base?

**Limitations:**

Yes

---

> ### Author Rebuttal · Authors · 2024-08-07
>
> Thank you for your review. We’re glad that you think that our metrics intuitively make sense and that the motivation for p-annealing is well thought out and presented.
>
> **Mismatch between figures and caption:** Thank you; this is a mistake and we will correct this in the paper. We have included the updated plots in the attached PDF. Further updates are outlined in our global response.
>
> **Less sparse SAEs have worse coverage:** Here are two related perspectives on the connection between sparsity and coverage:
> - BSPs are typically sparse (e.g. only a sparse subset of chess boards have a knight on E3). Thus, if the SAE features are dense (as is the case for high L0 SAEs), then they have no chance of corresponding to sparse BSPs; thus sufficiently dense SAEs must have low coverage scores.
> - Consistent with prior work, we find that sparser SAEs have more interpretable features. Since coverage is a measure of whether our SAEs have features belonging to a certain class of interpretable properties, more interpretable SAEs will also tend to have better coverage scores.
>
> (These two perspectives are really two sides of the same coin: the connection between sparsity and interpretable properties is a key motivator for applying SAEs for interpretability [1, 2].)
>
> **Separate plots for evaluating p-annealing:** In the three variable plots, we wanted to demonstrate that p-annealing is a Pareto improvement in L0 / Loss recovered over the standard SAE and at the same time showcase how our new metrics can help to distinguish improvements invisible to existing metrics. Specifically, it becomes difficult to distinguish between the gated and p-annealing SAEs using existing metrics, while their differences are still visible in plots using our new metrics. However, we agree that it is difficult to distinguish between the gated and p-anneal SAEs in these plots and will make sure to improve the presentation in the final version.
>
> **Efficacy of p-annealing:** Thank you for pointing this out. The final sentence in the conclusion should be updated to better reflect our results. Specifically, we find that standard SAEs trained using p-annealing consistently perform better than those trained with constant L1 penalty by existing proxy metrics and in terms of coverage. However, Gated SAEs are on par with them in terms of coverage. We will include a more nuanced discussion in the results section.
>
> **Qualitative analysis of trained SAEs:** We did perform qualitative analysis of the trained SAEs, e.g. Figure 1 includes examples of game states that correspond to interpretable SAE features that we found. However, we found many more interpretable features that did not make it into the paper, such as an “en passant” feature that only activates when an en passant capture is available. We will provide additional qualitative results of trained SAEs alongside the new metrics in the appendix.
>
> **F_low vs F_high distinction:** You're right, our split into low vs. high seemed ambiguous and subjective. To address this, we moved to a more principled division of BSPs into (1) board state BSPs, which correspond to the presence of a particular piece at a particular board space, and (2) researcher-selected strategy BSPs for chess This is described in more detail in the global rebuttal. Please find our revised figures in the attached PDF.
>
> **Board reconstruction without high precision features:** Only high precision features are used in calculating the metric. If there are no high precision features for a BSP, the board reconstruction score for that BSP would be 0. We will clarify this in the text.
>
> **Batch of train data:** We apologize for the confusion caused by our terminology. To clarify, we use a consistent dataset of 1000 games as our training set for identifying high-precision features across all Board State Properties (BSPs). An additional, separate set of 1000 games serves as our test set. We will clarify this in our updated paper.
>
> **Qualitative analysis of metrics:** In qualitative analysis of high level BSP features, the features are inherently interpretable because we screen for 95% precision. Thus, for example, at least 95% of a pin feature’s activations will be when there is a pin on the board. We also noticed that high level features tend to activate in specific scenarios. For example, a pin feature may only activate on a more specific type of pin, such as “a pin by a bishop in the bottom left corner of the board”.
>
> **Number of SAEs trained** The plots contain 40 SAEs from each SAE training methodology for each board game. We had additional sweeps of 40 Standard SAEs on layer 1 of each trained model, and on layers 1 and 6 of the randomly initialized models.
>
> **Line 220, optimal L0 range** We try to make the same point as in our comment on **Less sparse SAEs have worse coverage:** above. Thank you for pointing this out, the word optimal is too strong here. We will replace “optimal L0 range” with “expected L0 range in line with previous work”.
>
> **Correlation between proposed metrics and final SAE reconstruction loss**  While there's some ambiguity in the term "final SAE reconstruction loss”, we interpret this question as comparing `Loss Recovered` with our new metrics. Our metrics generally perform best at the elbow of the L0 / Loss Recovered plot, balancing sparsity and reconstruction quality. A higher Loss Recovered doesn't necessarily mean better performance on our metrics, as it often comes with higher L0 (less sparsity). Conversely, very low L0 tends to have low Loss Recovered. Thus, the relationship between our metrics and reconstruction loss isn't straightforward, as we're optimizing for a balance rather than maximizing either variable.
>
>
>
> References
>
> [1] A. Templeton et al., ‘Scaling Monosemanticity: Extracting Interpretable Features from Claude 3 Sonnet’, Transformer Circuits Thread, 2024.
>
> [2] Elhage, et al., "Toy Models of Superposition", Transformer Circuits Thread, 2022.

---

> > ### Comment · Reviewer_NNJv · 2024-08-12
> >
> > Thanks for your clarifications and updated figures. I have updated my score.

---

### Author Rebuttal · Authors · 2024-08-07

We have taken a number of steps to improve, simplify, and clarify our analyses:
- Multiple reviewers raised that the distinction between low-level and high-level BSPs seemed ambiguous and subjective. To address this, we moved to a more principled division of BSPs into (1) board state BSPs, which correspond to the presence of a particular piece at a particular board space (giving 8 x 8 x 12 board state BSPs for chess and 8 x 8 x 2 for Othello), and (2) researcher-selected strategy BSPs for chess, discussed in more detail below [1]. Our goal was to confine researcher choice to the strategy BSPs as much as possible, while keeping the class of board state BSPs natural and principled.
- We identified and fixed an error which resulted in ground-truth game boards being mislabeled in around 10% of our data. As a result, our corrected results are much more crisp.
- We trained a new sweep of SAEs to span a larger range of L0s, with an emphasis on SAEs with a low L0 (which tend to be more interpretable).

The Standard SAE w/ p-Annealing method yields a Pareto improvement over Standard SAEs, visible in both existing unsupervised metrics (Figures 1a, 1c, 2a, 2c) and our supervised metrics (Figures 1b, 1d, 2b, 2d, 3a, 3b). While Gated SAE, Gated SAE w/ p-annealing, and Standard w/ p-annealing are not clearly separable by unsupervised metrics (L0 and Fidelity), our supervised metrics based on board state BSPs (Figures 1b, 1d, 3a, 3b) clearly differentiate these approaches.

Please find our revised figures 1-3 in the attached PDF. Additionally, Figure 4 contains the experimental results from the Relative Reconstruction Bias metric suggested by reviewer pBfq. Our results indicate that p-annealing achieves a similar relative reconstruction bias compared to gated and gated w/ p-annealing, all of which are significantly improved over the standard SAE.

[1] Strategy BSPs were selected by the authors based on domain knowledge and prior chess model interpretability work [2] to be properties relevant to playing strategically in chess (for example BSPs that classify the presence of a pin or a fork on the board). Because the Othello model was trained to play random legal moves (rather than to play strategically), we do not consider strategy BSPs for Othello.

[2] T. McGrath et al., ‘Acquisition of chess knowledge in AlphaZero’, Proceedings of the National Academy of Sciences, vol. 119, no. 47, Nov. 2022.

---

### Comment · Area_Chair_9gfM · 2024-08-13
**Any final questions?**

Dear reviewers,
The discussion period is ending soon. If you have additional questions for the authors, please post them now.

One of the questions that two of the reviewers raised was the paper does not address enough domains, especially not natural language and so the utility of the approach may be harder to judge.

Reviewer pBfq, your rating differs significantly from the others -- have the author's responses addressed any of your concerns ?
Thanks.

---

### Decision · Program_Chairs · 2024-09-25

**Decision:**

Accept (poster)

**Comment:**

The paper proposes new ways to measure interpretability of features from SAEs in the setting of chess and othello based on known , interpretable BSP (board specific positions). The authors show that these metrics reveal things that normal reconstruction loss and L0 loss do not reveal; e.g. that larger SAE models are better than smaller SAE models on this. The authors also use an annealed training of SAE’s where  L^p_p Norm regularization is used and p is annealed from 1 to 0. The authors show that this results in more sparse features that are also better according to the metrics that the authors proposed.
The reviewers found the paper to be quite interesting. They did raise the concern that it is not clear how this setup might be useful for understanding interpretibility in normal language setting. The authors suggest that their p-annealing procedure produces gains that were not previously measurable and hope that such annealing procedures could be useful elsewhere.
The reviewers pointed out a couple of errors with the subfigures and captions in figure 2 and I hope the authors will fix this in the final version.